# OpenReview forum: "VALUEFLOW: Toward Pluralistic and Steerable Value-based Alignment in Large Language Models"
_ICLR.cc/2026/Conference — Submitted to ICLR 2026_

### Official Review · Reviewer_cFGL · 2025-10-21

**Soundness:** 3
**Presentation:** 2
**Contribution:** 3
**Rating:** 6
**Confidence:** 3

**Summary:**

This paper introduces VALUEFLOW, a unified framework for value-based LLM alignment focused on steerable intensity control. The authors argue that existing methods fail to capture value hierarchies or reliably measure value intensity, leading to unstable evaluations. VALUEFLOW addresses this with three components: HIVES, a hierarchical value embedding space; VIDB, a large-scale database of texts with intensity scores derived from robust ranking aggregation using a Plackett-Luce model; and a stable anchor-based evaluator that ranks outputs against VIDB. Using this framework, the authors conduct a comprehensive study on ten models across four value theories, characterizing model-specific asymmetries in steerability, identifying a "strong-anchor dominance" effect in multi-value control, and providing a scalable infrastructure for pluralistic alignment.

**Strengths:**

This paper addresses an important and timely question, how to systematically examine and steer the value representations embedded in current large language models under pluralistic and controllable conditions.

It proposes a comprehensive engineering framework that connects extraction, evaluation, and steering into a closed loop, forming an end-to-end pipeline for value-based alignment.

The experiments are large-scale and multi-dimensional, covering 10 models, 4 value theories, and 32 value dimensions, including both single-value and multi-value steering, refusal analysis, and downstream evaluation. These results provide broad and empirically grounded insights into asymmetric steerability and anchor dominance effects.

**Weaknesses:**

The reliance on LLM-as-a-Judge remains a concern. It would be valuable to include an analysis of inter-model differences in judging behavior, as well as quantitative consistency checks between LLM-based and human annotations.

The paper’s writing and structure are sometimes hard to follow; a clearer organization and smoother transitions would significantly improve readability.

There are two minor typos at Lines 296 and 417, the paragraph titles should be followed by a period.

**Questions:**

See the **Weaknesses** section.

---

> ### Author Response · Authors · 2025-11-23
>
> **Dear Reviewer cFGL,**
>
> Thank you for reviewing our submission and for the thoughtful feedback. Your comments have been very helpful in refining the paper.
>
> We respond to each point in the following.
>
> ---
>
> ### **W1. Reliability of ranking-based value evaluation**
>
> > The reliance on LLM-as-a-Judge remains a concern. It would be valuable to include an analysis of inter-model differences in judging behavior, as well as quantitative consistency checks between LLM-based and human annotations.
> >
>
> **Summary**:
>
> We conduct a comprehensive **human evaluation (2K scalar ratings, 1.5K ranking judgments)** and find **strong alignment with human judgment**s—**1.4** mean deviation, **85.3% pairwise accuracy, and 86.7% ±1-window accuracy**—demonstrating the reliability of our ranking-based evaluator.
>
> **Response**:
>
> Thank you for highlighting the concern regarding the use of LLM-as-a-Judge.
>
> While Section 3 provides initial evidence that our evaluator behaves robustly in the SVT setting, we agree that a more thorough validation is necessary to fully address reliability and inter-model variability.
>
> To this end, we conducted a **human evaluation study** covering all value theories used in our work. In total, we collected **2K scalar ratings** and **1.5K pairwise and windowed ranking judgments** from human annotators, enabling a direct comparison between LLM-based evaluations and human assessments.
>
> We assess alignment along three dimensions.
>
> - **VIDB score reliability.** We measure how closely the evaluator’s scalar intensity scores match human ratings by computing the mean absolute deviation across 2K annotated samples, and we benchmark performance against rating-based LLM baselines (Qwen3-32B, Phi-4, Gemma-3, Mistral-3.1) using a win-rate metric indicating how often a model’s score is closer to human judgment.
> - **Pairwise ranking accuracy.** For sampled text pairs, human annotators select which item better expresses a target value; we evaluate the evaluator’s agreement with these choices and compare the mean intensity gaps for consistent versus inconsistent cases to determine if disagreements arise from ambiguous comparisons.
> - **Windowed evaluation fidelity.** In a 6-window ranking setup, humans assign each text to one of six ordered “intensity windows,” and we measure exact-match accuracy, ±1-window accuracy, and mean positional deviation between human and evaluator assignments.
>
> **Table 1. VIDB score reliability.**
>
> | **Ours** | **VS. Qwen3** |  | **VS. Phi4** |  | **VS. Gemma3** |  | **VS. Mistral3.1** |  |
> | --- | --- | --- | --- | --- | --- | --- | --- | --- |
> | Avg. diff | Avg. diff | Win rate | Avg. diff | Win rate | Avg. diff | Win rate | Avg. diff | Win rate |
> | **1.4** | 2.1 | **60.4** | 4.2 | **66.5** | 2.5 | **65.5** | 4.2 | **78.7** |
>
> **Table 2. Pairwise and windowed evaluation.**
>
> | **Pairwise** |  |  | **Windowed** |  |  |
> | --- | --- | --- | --- | --- | --- |
> | Consistency (%) | Mean diff (consistent) | Mean diff (inconsistent) | Exact Acc | ±1 Acc | Mean diff |
> | **85.3** | 6.44 | 2.80 | **60.8** | **86.7** | 0.46 |
>
> Our evaluator consistently aligns with human judgments. For **VIDB score reliability**, it shows the lowest deviation from human scalar ratings (1.4) and outperforms all baselines with 60–79% win rates. For **pairwise ranking accuracy**, agreement with human selections reaches 85.3%, and the relatively small intensity gap for inconsistent cases (6.44 vs. 2.80) indicates that disagreements are concentrated in inherently ambiguous pairs. For **windowed evaluation fidelity**, the evaluator performs strongly even under the more difficult 6-window formulation, achieving 60.8% exact-match accuracy, 86.7% ±1-window accuracy, and a mean deviation below 0.5.
>
> These results demonstrate that our ranking-based evaluator is **highly consistent with human preferences**, more stable than rating-style LLM baselines, and robust across multiple value theories. We have added the results to **Section 7.2** and the detailed protocol to **Appendix G**. We have also added an analysis of inter-model differences in judgment behavior in **Appendix C.3** (VIDB construction) and **Appendix D.8** (generation-time evaluation).

---

> > ### Comment · Reviewer_cFGL · 2025-11-23
> >
> > Thank you for your reply. I'm curious about the number, backgrounds, and information distribution of the annotators you recruit for your human study, and I hope you can reveal more information in this regard.

---

> > > ### Author Response · Authors · 2025-11-24
> > >
> > > Thank you for your thoughtful question.
> > >
> > > Regarding the annotators, our study involved **25 participants**: 23 college graduates and 2 undergraduates. All were fluent in English, and we provided native-language value definitions when requested to ensure full comprehension. Participants were **volunteers drawn from university students and the broader public**, with all individuals providing **informed consent** prior to participation. They fell within the 20–59 age range, and we did not collect any sensitive personal data beyond education level and language proficiency. Before beginning the tasks, all participants completed a tutorial covering the value theories and definitions.
> > >
> > > Regarding **task distribution**, each annotator completed approximately 80 score-reliability items and 60 ranking-accuracy items (formats detailed in Appendix G, Figures 42 & 43). To ensure balanced coverage, each annotator was assigned four specific values (20 score items and 15 ranking items per value). Items were sampled randomly, and annotators were blinded to others' responses.
> > >
> > > We will add these demographic, recruitment, and distributional details—as well as per-value statistics—to **Appendix G** in the revised paper. We also plan to release the anonymized human-evaluation subset upon publication.
> > >
> > > **Please let us know if any additional clarification would be helpful.**

---

> > > > ### Comment · Reviewer_cFGL · 2025-11-24
> > > >
> > > > Thank you for your reply. I will maintain my score.

---

> > > > > ### Author Response · Authors · 2025-11-26
> > > > >
> > > > > Thank you for your follow-up message. We greatly appreciate the time you have taken to review our work and for maintaining your score after the clarifications. Your comments throughout the discussion have been very helpful in improving both the clarity and the rigor of the paper. We will carefully reflect your suggestions in the camera-ready revision, including clearer presentation and expanded human-evaluation details.
> > > > >
> > > > > Thank you again for your thoughtful engagement with our submission.

---

> ### Author Response · Authors · 2025-11-23
>
> ### **W2. Readability**
>
> > The paper’s writing and structure are sometimes hard to follow; a clearer organization and smoother transitions would significantly improve readability.
> >
>
> > There are two minor typos at Lines 296 and 417, the paragraph titles should be followed by a period.
> >
>
> Thank you for the helpful suggestion.
>
> We acknowledge that, due to space constraints, some details and transitions were compressed in the current draft. As several reviewers have noted, certain design motivations and the roles of individual components (e.g., why each part of the framework is required and how they connect) were not sufficiently emphasized. In the revision, we will substantially clarify the organization of the paper—adding clearer section-level roadmaps, improving transitions between extraction, evaluation, and steering, and expanding the motivation behind each design choice. We believe these revisions will enhance readability and make the overall narrative smoother and more coherent.
>
> Thank you as well for catching the typos at Lines 296 and 417. We have corrected these paragraph titles to include the required periods.
>
> ---
>
> **Thank you for your constructive comments. We hope the updated results respond adequately to your concerns.**

---

### Official Review · Reviewer_wGtp · 2025-10-25

**Soundness:** 3
**Presentation:** 3
**Contribution:** 3
**Rating:** 6
**Confidence:** 3

**Summary:**

This paper introduces VALUEFLOW, a unified framework for value-based alignment in large language models, encompassing three core components: extraction, evaluation, and intensity-controlled steering. The framework includes: (1) HIVES - a hierarchical value embedding space capturing multi-theory value structures; (2) VIDB - a large-scale value intensity database with calibrated intensity estimates via ranking aggregation; (3) an anchor-based evaluator producing consistent intensity scores through ranking rather than rating. The authors conduct large-scale experiments across 10 models and 4 value theories, identifying asymmetries in steerability and composition laws for multi-value control, while demonstrating improved demographic alignment through value-based profiling.

**Strengths:**

1. Problem Focus and Contribution Scope:
    - Addresses three critical gaps in value alignment research: extraction lacks hierarchical structure, evaluation detects presence but not intensity, and steerability remains insufficiently understood - the problem formulation is clear and important.
    - First to propose "steerability with intensity," extending value alignment from directional control to graded intensity control, opening a new dimension for pluralistic value alignment.

2. Method Mechanism:
HIVES's two-stage training design is well-motivated: Stage 1 aligns intra-theory structure via hierarchical contrastive learning, while Stage 2 unifies heterogeneous theories through cross-theory anchors - the technical approach is sound.

3. Empirical Evaluation:

    - Comprehensive experimental scope: 10 mainstream models (open and closed source) × 4 value theories × 32 value dimensions × 500 prompts, providing broad coverage.
    - Valuable empirical patterns discovered: asymmetry in negative steering, strong-anchor dominance effect, value similarity affecting composition laws, etc.

**Weaknesses:**

- Technical Correctness: Plackett-Luce model assumes Independence of Irrelevant Alternatives (IIA), but value judgments may exhibit context-dependent effects; the paper does not discuss robustness when this assumption is violated.

- Evaluation Scope: Experiments mainly focus on "short-term prompt-driven value steering" and do not explore the stability of value expression in long-term dialogues (e.g., whether the model deviates from the target intensity after multi-turn interactions). They also fail to test value alignment effects in low-resource language or niche cultural contexts, resulting in limited scenario coverage.

- Loss function weights in two-stage training (λind=0.5, λtheory=1.0) lack ablation study support; the impact of different weight configurations on final representation quality is unknown.

**Questions:**

- Theoretical Foundation: Why is the Plackett-Luce model suitable for value intensity aggregation? Have other ranking models (e.g., Bradley-Terry, Thurstone) been tested? How does the model perform when value judgments exhibit non-transitivity?

- How do the 274 cross-theory anchor concepts ensure balanced coverage across theories? Is there any theory dominating the anchor set?

- How applicable is the framework to non-English languages or non-Western value systems (e.g., Confucian, Buddhist values)?

- Is the computational cost of ranking evaluation at inference time (k=6, m=3 implies 18 LLM calls) acceptable in real applications?

---

> ### Author Response · Authors · 2025-11-23
>
> **Dear Reviewer wGtp,**
>
> Thank you for your careful and constructive review. We appreciate the time you spent evaluating our work and the helpful suggestions you provided.
>
> We address your points below.
>
> ### **W1 & Q1.  Theoretical justification for Plackett–Luce model**
>
> > Technical Correctness: Plackett-Luce model assumes Independence of Irrelevant Alternatives (IIA), but value judgments may exhibit context-dependent effects; the paper does not discuss robustness when this assumption is violated.
> >
>
> > Theoretical Foundation: Why is the Plackett-Luce model suitable for value intensity aggregation? Have other ranking models (e.g., Bradley-Terry, Thurstone) been tested? How does the model perform when value judgments exhibit non-transitivity?
> >
>
> **Summary:**
>
> The **Plackett–Luce** family is well suited to our setting because it absorbs noisy, context-dependent, and occasionally cyclic comparisons into stable continuous utility estimates rather than enforcing strict transitivity. Empirically, PL with moderate window sizes **converges far faster** than Bradley–Terry or Thurstone, motivating its use for both  efficient real-time evaluation.
>
> **Response:**
>
> Thank you for raising these important questions regarding the theoretical justification of the Plackett–Luce (PL) model.
>
> Our objective is to recover a **continuous latent intensity score** from noisy, context-dependent comparisons—not to enforce perfect global transitivity. Human and LLM judgments often contain mild inconsistencies or cycles, especially among texts with **similar intensity**, and probabilistic models like Plackett–Luce (PL) naturally smooth these by pulling the corresponding utilities closer together.
>
> Although PL is derived under IIA, **our use does not rely on IIA holding exactly**. With many heterogeneous comparisons, IIA violations behave like stochastic noise, and PL is well known to act as a **regularized MLE** that yields stable latent scales even when comparisons are partially inconsistent. This smoothing effect is precisely what we need: across overlapping ranking windows, contextual variation averages out, producing a robust intensity axis.
>
> Building on this perspective, we also justify the use of PL for both **VIDB construction** and **evaluation**.
>
> **(1) VIDB Construction**
>
> VIDB involves aggregating ~300K comparisons per value obtained from multiple LLMs and sampling strategies. As shown in Appendix C.3, we found that using **PL with window size w=2**—which reduces to the **Bradley–Terry (BT)** model—yields the most stable aggregation at this scale. Additionally, in near-tie or cyclic regions, BT naturally assigns **similar utilities**, which is precisely what we want. This redundancy across many heterogeneous comparisons helps diminish the practical impact of local IIA violations or non-transitivities and yields smooth utility curves in practice.
>
> **(2) Evaluation Phase**
>
> In the evaluation phase, we must jointly consider **stability**, **comparative accuracy**, and **computational efficiency**. Each ranking window requires a full model call, and modern LLM inference is dominated by the prefill phase, especially when decoding is short. Thus, minimizing the number of ranking windows (**m**) is critical for making the evaluator usable in real time. To further reduce variance and mitigate context effects, we adopt **bucketed sampling** as the default strategy: for each window, we select k−1 anchor texts by stratifying the VIDB into intensity buckets spanning [−10,10]. This ensures that each window contains a balanced and representative spread of intensities, yielding more informative comparisons than uniform random sampling while avoiding the bias that can arise from anchor panels.
>
> We evaluated three latent-utility models under equal comparison budgets m:
>
> - Bradley–Terry (w = 2)
> - Thurstone (w = 2)
> - Plackett–Luce (w = 6)
>
> Using **w = 30** as a near-converged reference point, we computed the mean absolute deviation of inferred scores for each method at varying numbers of windows **m**:
>
> **Table 1: Ablation on convergence behaviors**
>
> | m | BT (w=2) | Thurstone (w=2) | PL (w=6) |
> | --- | --- | --- | --- |
> | 1 | 2.7 | 2.5 | **1.5** |
> | 2 | 2.0 | 1.9 | **1.1** |
> | 3 | 1.7 | 1.5 | **0.8** |
> | 5 | 1.3 | 1.2 | **0.6** |
> | 10 | 0.8 | 0.7 | **0.3** |
> | 20 | 0.3 | 0.2 | **0.1** |
> | 30 | 0.0 | 0.0 | 0.0 |
>
> As shown, **PL with w=6 converges substantially faster** than BT or Thurstone for the same number of model calls. Because evaluation must run in real time, we adopt **m = 3** windows by default. Under this budget, PL-6 yields **< 1-point deviation** on a 20-point scale (< 5% relative error), which we consider sufficiently stable for practical evaluation.
>
> We have included the full convergence plots and sensitivity analyses for both **window size (k)** and **number of windows (m)** in **Appendix D.8**, which visualize this rapid convergence and support our choice of PL-6 (with bucketed sampling) as the default evaluator.

---

> ### Author Response · Authors · 2025-11-23
>
> ### **W2.1. Multi-turn Evaluation**
>
> > Experiments mainly focus on "short-term prompt-driven value steering" and do not explore the stability of value expression in long-term dialogues (e.g., whether the model deviates from the target intensity after multi-turn interactions).
> >
>
> **Summary:**
>
> We also analyze multi-turn behavior and observe a gradual degradation of injected value intensity across turns, confirming that value steering weakens in long-horizon dialogues and demonstrating the flexibility of our evaluator for tracking such consistency dynamics.
>
> **Response:**
>
> We thank the reviewer for raising this point. While long-horizon consistency was not our primary objective, we did not include such experiments in the main evaluation and left an in-depth analysis to future work.
>
> However, our framework can naturally be extended to study consistency. To demonstrate this, we conducted an additional **100-turn dialogue evaluation** using the GPV questionnaire. We inject the target intensity **only in the first turn**, then randomly shuffle and ask the questionnaire for 100 turns, repeating the entire process **50 times** for robustness. Each turn’s response is evaluated independently using the same evaluator protocol used in the main paper.
>
> Across values and models, we observe a clear **diminishing effect** of injected values over turns. For example, in *benevolence*, both strong opposition (−2) and strong support (+2) gradually drift toward neutrality:
>
> **Table 2: Multi-turn intensity change**
>
> | intensity | turns | mean score |
> | --- | --- | --- |
> | −2 | 0–20 | −3.94 |
> | −2 | 20–40 | −2.95 |
> | −2 | 40–60 | −3.19 |
> | −2 | 60–80 | −2.86 |
> | −2 | 80–100 | −2.74 |
> | +2 | 0–20 | 3.68 |
> | +2 | 20–40 | 3.56 |
> | +2 | 40–60 | 3.61 |
> | +2 | 60–80 | 3.36 |
> | +2 | 80–100 | 3.42 |
>
> Negative injections tend to decay slightly more rapidly than positive ones, which aligns with prior observations that **value-consistent behavior attenuates as conversational context grows**. This supports the interpretability and utility of our evaluator for analyzing consistency over time.
>
> We added these results in **Appendix D.9**.
>
> ---
>
> ### **W2.2, Q3. Value alignment for low-resource language and cultural contexts**
>
> > How applicable is the framework to non-English languages or non-Western value systems (e.g., Confucian, Buddhist values)?
> >
>
> **Summary:**
> We also show that our framework generalizes to **new languages** and **non-Western value theories**, constructing tailored corpora and value items and demonstrating that the full VIDB–evaluation pipeline remains effective. This illustrates the framework’s broad adaptability beyond English settings.
>
> **Response:**
>
> We thank the reviewer for highlighting this issue. Our initial focus on English and widely used value theories (e.g., Schwartz, MFT) reflects the practical constraints of existing resources: (i) most value-related corpora and benchmarks are developed in English, and (ii) English dominates publicly available web text (over 70%, Hasman 2009), making it the only language where large-scale value-eliciting datasets are readily accessible. Culturally specific value systems (e.g., Confucian, Buddhist ethics) likewise suffer from a lack of standardized value inventories or curated corpora.
>
> However, our framework is **not** limited to English or Western value theories. We intentionally design a **lightweight and theory-agnostic extension pipeline** that enables the construction of both (i) *language-specific value corpora* and (ii) *domain- or culture-specific value systems*.
>
> For languages, we collect 10K documents for each target language (e.g. Arabic, Chinese, Korean) from the CulturaX corpus, apply filtering to isolate naturally occurring text, and extract value-eliciting segments aligned with the Schwartz values. For new value systems where standardized inventories do not exist, we build a domain corpus from relevant communities (e.g., the Buddhism subreddit) and process it to **derive a set of culturally distinctive value items (**e.g. *mindfulness and karma from Buddhism)*. These items are curated and deduplicated using LLM-assisted refinement.
>
> Using these language- and domain-specific value items, we then **construct full value-intensity databases** following the same VIDB protocol used for English, and we further evaluate **steerability** under these new settings.
>
> We include the full procedure and resulting analyses in **Section 7.2 (Fig. 10)** and **Appendix F**, demonstrating that our framework generalizes robustly to non-English languages and culturally specific value systems.

---

> ### Author Response · Authors · 2025-11-23
>
> ### **W3. Ablation on loss coefficients in HiVES training**
>
> > Loss function weights in two-stage training (λind=0.5, λtheory=1.0) lack ablation study support; the impact of different weight configurations on final representation quality is unknown.
> >
>
> **Summary:**
>
> We select the loss coefficients that best balance ranking accuracy and cross-theory stability, with (lambda_indiv = 0.5, lambda_theory = 1.0) providing the strongest overall performance.
>
> **Response:**
>
> We value this comment and provide clarification below. We conducted an ablation over Stage-2 loss weights using a grid search at an early checkpoint (~10k steps, ≈20% of full training) over:$$\lambda_{\text{indiv}} \in \{0, 0.5, 1.0\}, \qquad \lambda_{\text{theory}} \in \{0, 0.5, 1.0, 1.5, 2.0\}$$
> We evaluated each setting using three criteria:
> 1. Pairwise ranking accuracy (hierarchical alignment quality).
> 2. Anchor gap:
> $$\Delta_{\text{anchor}} = \mathbb{E}[d(x,y) \mid \text{same anchor}] - \mathbb{E}[d(x,y) \mid \text{different anchor}]$$
> 3. Individual gap:
> $$\Delta_{\text{indiv}} = \mathbb{E}[d(x,y) \mid \text{same indiv}] - \mathbb{E}[d(x,y) \mid \text{different indiv}]$$
>
> **Table 3: Ablation on loss coefficients**
>
> | λ_indiv | λ_theory | Pairwise Rank Acc | Anchor Gap | Indiv Gap |
> | --- | --- | --- | --- | --- |
> | 0.0 | 0.5 | 0.62 | 0.03 | 0.01 |
> | 0.0 | 1.0 | 0.61 | 0.07 | 0.01 |
> | 0.0 | 1.5 | 0.61 | 0.07 | 0.01 |
> | 0.0 | 2.0 | 0.58 | 0.07 | 0.01 |
> | 0.5 | 0.0 | 0.64 | 0.03 | 0.03 |
> | 0.5 | 0.5 | 0.64 | 0.04 | 0.03 |
> | **0.5** | **1.0** | **0.65** | **0.11** | **0.03** |
> | 0.5 | 1.5 | 0.61 | 0.11 | 0.03 |
> | 0.5 | 2.0 | 0.60 | 0.07 | 0.01 |
> | 1.0 | 0.0 | 0.62 | 0.06 | 0.03 |
> | 1.0 | 0.5 | 0.62 | 0.08 | 0.03 |
> | 1.0 | 1.0 | 0.61 | 0.05 | 0.02 |
> | 1.0 | 1.5 | 0.60 | 0.05 | 0.02 |
> | 1.0 | 2.0 | 0.59 | 0.07 | 0.03 |
>
> Most gap values were small across the grid, indicating stable cross-theory behavior. The configuration $(\lambda_{\text{indiv}}=0.5, \lambda_{\text{theory}}=1.0)$ yielded the strongest ranking accuracy while keeping both gap measures controlled, and was therefore selected for full training.
>
> We will include this ablation and expanded explanation in the revised appendix.
>
> ---
>
> ### **Q2. Effect of cross-theory anchors**
>
> > How do the 274 cross-theory anchor concepts ensure balanced coverage across theories? Is there any theory dominating the anchor set?
> >
>
> Thank you for the question.
>
> As summarized in **Appendix B.3**, our construction pipeline includes filtering steps that remove low-support clusters and merge near-duplicates to prevent any single theory from dominating the anchor space. To further verify balance, we measured how each anchor distributes across the four theories by computing normalized per-theory proportions within each anchor and then averaging these across all 274 anchors. The resulting mean distribution—Duties **23.6%**, MFT **25.6%**, Schwartz **27.7%**, Rights **22.1%**—is tightly clustered around 25%, indicating that **no theory dominates and that the anchors provide balanced**, cross-theory representation as intended. We have included this results in **Section 4.2.**
>
> ---
>
> ### **Q4. Computational cost**
>
> > Is the computational cost of ranking evaluation at inference time (k=6, m=3 implies 18 LLM calls) acceptable in real applications?
> >
>
> Thank you for raising this practical concern.
>
> In our implementation, the window size k represents the number of anchor candidates evaluated **within a single LLM call**, since the model ranks all k anchors jointly in one prompt. The repetition factor m controls how many times this ranking is repeated for robustness. Therefore, with k=6 and m=3, the evaluator requires only **3** short LLM calls. Each call generates a brief ranking output over a small anchor set (roughly ~10 tokens), and compared to the substantially larger cost of producing full model responses in real applications—often hundreds of tokens—the overhead of these 3 lightweight calls is negligible.
>
> We will make this more explicit in the revised script.
>
> ---
>
> **We hope these findings resolve your concerns. Thank you again for your helpful feedback.**

---

> > ### Comment · Reviewer_wGtp · 2025-11-27
> >
> > Thanks for the explanation, which clears my concerns. I have no other concerns.

---

> > > ### Author Response · Authors · 2025-11-27
> > >
> > > Thank you for your continued engagement with our submission and for letting us know that your concerns have been resolved. We appreciate the thoughtful questions you raised throughout the review process, as they helped us clarify several methodological details and improve the overall presentation of the framework.
> > >
> > > In the revision, we will incorporate the relevant explanations from the rebuttal into the main text and the appendix, particularly those related to the PL-model justification, cross-theory anchors, and evaluation design.
> > >
> > > Thank you again for your constructive feedback and for helping us strengthen the clarity and completeness of the paper.

---

### Official Review · Reviewer_Uk4Z · 2025-10-31

**Soundness:** 2
**Presentation:** 2
**Contribution:** 3
**Rating:** 2
**Confidence:** 3

**Summary:**

Beyond preference-based methods, this paper accounts for alignment with real human values that are diverse and serve as more stable principles in decision-making.

It aims to address two main challenges: (1) value extraction, current studies rely on static questionnaires or simple judgments, limiting the ability to capture signals from open-ended conversational contexts and rarely encode the hierarchical nature of values. (2) value evaluation, current studies measure presence rather than strength, overlooking intensity in open-ended outputs.

For these, they introduce a unified framework VAUEFLOW, with HIVES, a hierarchical value embedding space to capture value profiles, and a ranking-based value evaluator together with a value-intensity database VIDB. Then, they conduct experiments to verify the effectiveness of HIVES and the ranking-based evaluator, as well as the whole framework for value steerability.

**Strengths:**

1. This paper proposes VALUEFLOW, a unified framework spanning value extraction, evaluation and steering in LLMs, allowing for end-to-end steerable value alignment.
2. To address the challenge of value extraction, it constructs a HIVES method to unify heterogeneous value theories.
3. Accounting for the intensity of values and instability of current rating-based evaluations, this paper builds a value-intensity database and designs a ranking-based evaluation method of intensity.
4. Some experiments and analysis are conducted to demonstrate the usage in value steerability.

**Weaknesses:**

1. Baselines of value extraction on open-ended conversational contexts are largely ignored both in the Introduction part (Line 52) and Related Work part. I think there are some works on this task.
2. The whole method needs better clarification:
- A structural algorithm is desired to formulate the whole framework, especially how the hierarchical value embedding space is built.
- More descriptions are required for the Sec 4.3 Two Stage Training Process, what are the inputs and what are the outputs? How to obtain the ground truth data for training?
- What is the value steering method for AI used in this paper?
- What are the ground truth and evaluation metrics used for the experiments in Figure 4.
3.  There lack sufficient baselines for comparison in both Table 1 and Table 2, limiting the reliability of effectiveness about your method. There are evaluators in Denevil, ValuePrism, etcs mentioned in this paper, which should be considered for comparison.
4. There are some problematic settings in your method and experiments.
- In Line 259, when constructing the VIDB dataset, you use multiple LLMs to generate the intensity rating label for each text. However, you mentioned in Sec 3.2 that LLMs’ ratings on value intensity are highly unstable. So this would decrease the accuracy of the VIDB dataset.
- In Sec 6.3, you first construct value profiles from the dataset, then use these value profiles as the alignment target, finally compute the accuracy between the alignment with the data which are used for constructing value profiles. This could incur a risk of data contamination, limiting the significance of the experiments.

**Questions:**

1. Figure 2 is currently a little confusing about which LLM generates the rating score respectively.
2. Figure 8 is hard to understand.

---

> ### Author Response · Authors · 2025-11-23
>
> **Dear Reviewer Uk4Z,**
>
> Thank you for taking the time to provide a detailed and thoughtful review. We appreciate your comments on both the strengths of our framework and the areas needing clearer explanation or additional comparison. Your feedback has been very helpful for improving the clarity and rigor of the paper.
>
> We address each point in detail below.
>
> ---
>
> ### **W1. Baselines of value extraction on open-ended contexts**
>
> > 1. Baselines of value extraction on open-ended conversational contexts are largely ignored both in the Introduction part (Line 52) and Related Work part. I think there are some works on this task.
> >
>
> Thank you for the helpful question.
>
> We acknowledge that several prior works have indeed addressed value extraction in open-ended or conversational contexts, such as [1], [2], [3], [4], [5]. While these studies were previously summarized in **Appendix A.2**, we agree that they deserve greater emphasis in the main text.
>
> Accordingly, we have revised the ***Related Work*** section to more explicitly discuss these open-ended value extraction baselines.
>
> [1] Kovac, M., et al. *Stick to Your Role! Stability of Personal Values Expressed in Large Language Models.* PLOS ONE, 2024.
>
> [2] Jiang, Z., et al. *Raising the Bar: Investigating the Values of Large Language Models via Generative Evolving Testing.* In Proceedings of ICML, 2025.
>
> [3] Klingefjord, O., et al. *What Are Human Values, and How Do We Align AI to Them?* arXiv preprint, 2024.
>
> [4] Huang, X., et al. *Values in the Wild: Discovering and Analyzing Values in Real-World Language Model Interactions.* arXiv preprint, 2025.
>
> [5] Mirzakhmedova, N., et al. *The Touché23-ValueEval Dataset for Identifying Human Values Behind Arguments.* arXiv preprint, 2023.

---

> ### Author Response · Authors · 2025-11-23
>
> ### **W2.1. A structural algorithm for the whole framework**
>
> > A structural algorithm is desired to formulate the whole framework, especially how the hierarchical value embedding space is built.
> >
>
> Thanks for the suggestion. We have added a complete structural formulation of our framework. Specifically, we now include **Algorithm 1–3** in **Appendix B.5**, which detail the full procedure for constructing the embedding space. In addition, we provide **Algorithm 4** in **Appendix D.1**, which formalizes the value-intensity evaluation protocol used in our experiments.
>
> ---
>
> ### **W2.2. Clarification on Two Stage Training Process**
>
> > More descriptions are required for the Sec 4.3 Two Stage Training Process, what are the inputs and what are the outputs? How to obtain the ground truth data for training?
> >
>
> **Summary**:
> The two-stage process takes **texts with mapped hierarchical value labels** as ground-truth inputs, learns **theory-consistent embeddings** in Stage 1, aligns them to **cross-theory anchors** in Stage 2, and outputs a **unified value embedding space (HIVES)**.
>
> **Response**:
>
> Thank you for the helpful comment.
>
> We agree that Section 4.3 was concise due to page limits, and that this may have obscured the end-to-end data flow. We have added a detailed structural algorithm to **Appendix B (Algorithm 3)** to clarify the full procedure.
>
> The **ground truth** comes from the mapping procedure in Sections 4.1–4.2, which assigns each text a **hierarchical value path** (e.g., *SVT → Benevolence → Caring*) and a **direction label** (*supports / opposes / not related*). These paired (text, label-path) tuples serve as the training inputs.
>
> - **Stage 1 (Intra-theory alignment):**
>
>     Using only labels within each theory, the model learns hierarchy-consistent embeddings via a hierarchical contrastive loss, treating texts with shared prefixes and direction as positives and others as negatives.
>
> - **Stage 2 (Inter-theory alignment):**
>
>     The text embeddings are then aligned to the cross-theory anchors and value instances constructed in Section 4.2. These anchors serve as semantic prototypes, providing supervision to unify heterogeneous value theories.
>
>
> We will also add an explanation to **Section 4.3** summarizing these relationships and training signals for clearer readability.
>
> ---
>
> ### **W2.3. Value steering method of the paper**
>
> > What is the value steering method for AI used in this paper?
> >
>
> **Summary**:
> We focus on **prompt-based steerability** (intensity-anchor and user-text prompting), supplemented by **profile-based inputs** and **additional activation-based steering.**
>
> **Response**:
>
> Thank you for the question. We appreciate the opportunity to clarify this point.
>
> As described in **Section 6.2 (line 357-375)**, we primarily adopt **two prompting-based steering methods**:
>
> (1) **Intensity-anchor prompting**, which extends value–anchor prompts with explicit intensity cues (e.g., “+2: strongly values,” “−2: strongly rejects”), and
>
> (2) **User-text prompting**, which uses representative examples from our Value Intensity Database (VIDB) corresponding to different intensity levels.
>
> To complement these prompt-based approaches, we also analyze **non-prompt–based steering (described in line 522-526)** methods such as **activation-based** and **embedding-conditioned** steering (**Appendix D.5**), which showed limited but measurable control effects.
>
> ---
>
> ### **W2.4. Ground truth and evaluation metrics used for Figure 4**
>
> > What are the ground truth and evaluation metrics used for the experiments in Figure 4.
> >
>
> **Summary**:
> Figure 4 evaluates embeddings using **hierarchical ranking accuracy**, **similarity–label correlation**, and **value-vector orthogonality**, all computed against the **ground-truth hierarchical value paths and direction labels** derived from our unified annotation pipeline.
>
> **Response**:
>
> Thank you for the helpful question. We agree that the original description was brief, and we will revise it for clarity.
>
> As described in **Section 6.1** and **Appendix B.6**, our ground truth is derived from hierarchical value annotations: each text is assigned a multi-level theory path together with a directional tag. These labels define the pairwise similarity ordering used for both training and evaluation.
>
> For Figure 4, we report the three metrics:
>
> - **Hierarchical Ranking Accuracy** — evaluates whether cosine similarities follow the expected ordering implied by hierarchical distance
> - **Similarity–Label Correlation** — measures the correlation between cosine similarity and a numeric affinity score derived from shared value levels and directional agreement.
> - **Value-Vector Orthogonality** — assesses how close the value axes are to orthogonal, indicating disentanglement across value dimensions.
>
> Comprehensive definitions and formal equations for all metrics are provided in **Appendix B.6.**

---

> ### Author Response · Authors · 2025-11-23
>
> ### **W3. Baselines for Table 1 and the Table 2**
>
> > There lack sufficient baselines for comparison in both Table 1 and Table 2, limiting the reliability of effectiveness about your method. There are evaluators in Denevil, ValuePrism, etcs mentioned in this paper, which should be considered for comparison.
> >
>
> **Summary**:
>
> Since existing evaluators provide **categorical rather than graded signals,** we did not include them as baselines. Instead, we use the ValueBench rating evaluator as the appropriate comparison for continuous intensity evaluation.
>
> **Response**:
>
> Thank you for this thoughtful comment. We are grateful for the chance to elaborate on this aspect.
>
> We did not include evaluators such as **Value-Kaleido (ValuePrism)** or the **ValueEval classifier** because these models are primarily designed as **categorical detectors**. Their outputs effectively capture whether a value is present but are not intended to provide a calibrated measure of **how strongly** the value is expressed. Since our evaluation focuses on *continuous value intensity*, we used the **rating-based evaluation protocol** from ValueBench as the most comparable baseline.
>
> To further clarify this distinction, we conducted an additional experiment to examine whether classifier probabilities might still reflect graded intensity. For each value, we generated six levels of progressively stronger statements (0 → 5). For example, for **Tradition**:
>
> 1. “I feel neutral about traditions; I can follow them or not depending on the situation.”
> 2. “I somewhat appreciate traditions and try to follow them when I can.”
> 3. “I generally value traditions and make an effort to follow them when possible.”
> 4. “I value traditions and try to honor them regularly.”
> 5. “I strongly value traditions and feel a clear responsibility to uphold them.”
> 6. “I deeply value traditions and feel committed to preserving them for future generations.”
>
> Across 100 such sweeps, the average classifier support probabilities were:
>
> | Intensity | ValueEval | Value-Kaleido-XL |
> | --- | --- | --- |
> | 0 | 0.183 | 0.061 |
> | 1 | 0.995 | 0.996 |
> | 2 | 0.996 | 0.998 |
> | 3 | 0.987 | 0.995 |
> | 4 | 0.993 | 0.993 |
> | 5 | 0.993 | 0.991 |
>
> We observed that the probabilities quickly saturate after the first non-neutral level and remain nearly unchanged afterward. This suggests that, in practice, classifier outputs may not provide a reliably *graded* or *monotonic* signal aligned with intensity—likely reflecting their design focus on **presence detection** rather than **fine-grained strength estimation**.
>
> In light of this, classifier-based evaluators may be less suitable for the specific goal of continuous intensity measurement. Our **ranking-based evaluator** in VALUEFLOW was developed precisely to complement this gap: it provides **relative, fine-grained, and theory-consistent** intensity estimates that remain sensitive across subtle semantic differences.
>
> We appreciate the opportunity to clarify this point and will update the manuscript accordingly.
>
> ---
>
> ### **W4.1. Rating VS. Ranking measure for VIDB construction**
>
> > There are some problematic settings in your method and experiments.
> >
> > - In Line 259, when constructing the VIDB dataset, you use multiple LLMs to generate the intensity rating label for each text. However, you mentioned in Sec 3.2 that LLMs’ ratings on value intensity are highly unstable. So this would decrease the accuracy of the VIDB dataset.
>
> **Summary**:
>
> Indeed, VIDB is constructed using **ranking-based judgments**—avoiding the instability inherent in rating-based measures—and is therefore designed to provide a more reliable foundation for value-intensity estimation.
>
> **Response:**
>
> Thank you very much for raising this point. We apologize for the ambiguous wording around Line 259, which may have suggested that we rely on **absolute rating-style labels** when constructing the VIDB. We will revise the text to make this distinction clearer.
>
> To clarify: although Section 3.2 shows that **rating-based** intensity scores from LLMs can be unstable, the **VIDB construction** **does not** **use absolute ratings** at any stage. As detailed in Section 4, VIDB is built entirely from **relative, ranking-based comparisons**—LLMs only decide which of two (or more) texts expresses the value more strongly. These *pairwise and windowed preferences* are empirically much more stable across prompts and models.
>
> Furthermore, to mitigate noise from any single comparison, we aggregate all preferences through **Plackett–Luce optimization**, which produces a globally consistent latent intensity ordering.
>
> We appreciate the opportunity to clarify this, and we will revise the manuscript to make the ranking-based nature of the VIDB construction more transparent.

---

> ### Author Response · Authors · 2025-11-23
>
> ### **W4.2. Value profile construction from dataset**
>
> > In Sec 6.3, you first construct value profiles from the dataset, then use these value profiles as the alignment target, finally compute the accuracy between the alignment with the data which are used for constructing value profiles. This could incur a risk of data contamination, limiting the significance of the experiments.
> >
>
> **Summary**:
>
> Although profile construction and evaluation are **often required to draw from the same dataset** in this setting, we **minimize overlap through a disjoint-split protocol**—using only about **5%** of the data for profile estimation and ensuring none of those instances are included in the alignment accuracy evaluation.
>
> **Response:**
>
> Thank you very much for raising this important point.
>
> We understand the concern that constructing value profiles and evaluating alignment from the same dataset may create a risk of data contamination. In practice, however, this setup is **practically inevitable in this setting**, as population-level value profiles require text distributions that are rarely available from independent external sources (e.g., value-eliciting texts from very specific subpopulations such as *Northeast U.S. respondents*). For this reason, prior work—including OpinionQA studies [1–4]—similarly derives group-level profiles from a small subset of the available data and evaluates on the remainder.
>
> Following this established protocol, we **use only about 5% of the attribute-conditioned data** to estimate each value profile, which is significantly smaller than the 10–50% commonly used in prior literature. Importantly, **none of the instances used for profile estimation are reused during the alignment evaluation**, ensuring a clean separation between construction and testing.
>
> We will revise **Section 6.3** to describe this disjoint-split procedure more clearly.
>
> [1] Zhao et al., *Group Preference Optimization: Few-Shot Alignment of Large Language Models*, ICLR 2024.
>
> [2] Hwang et al., *Aligning Language Models to User Opinions*, Findings of EMNLP 2023.
>
> [3] Xuan Long et al., *Aligning Large Language Models with Human Opinions through Persona Selection and Value–Belief–Norm Reasoning*, COLING 2025.
>
> [4] Sorensen et al., *Value Profiles for Encoding Human Variation*, EMNLP 2025.
>
> ---
>
> ### **Q1 & Q2:**
>
> > Figure 2 is currently a little confusing about which LLM generates the rating score respectively. Figure 8 is hard to understand.
> >
>
> Thank you for pointing this out. As you have mentioned, Figures 2 and 8 may appear ambiguous in their current form.
>
> **For Figure 2**, we will explicitly label which LLM corresponds to each rating score (e.g., “Gemma-3: −8”, “Phi-4: +10”, “Mistral-3.1: −10”, “Qwen-3: 0”). We will also adjust the layout by drawing arrows or connectors from each model icon directly to its rating bar so that readers can clearly see which model produced which score.
>
> **For Figure 8**, we agree that the visualization requires clearer explanation. The **left panel** shows *two-value steering*, where two Schwartz values (e.g., Benevolence–Universalism, or Hedonism–Conformity) are simultaneously conditioned with intensity levels I∈{−2,−1,+1,+2} — where +2 denotes strong positive and −2 strong negative steering. The arrows indicate the *steering gain* Delta = s_steered - s_default along both axes.
>
> The **right panel** extends this to *five-value steering*, where each value receives its own input intensity (the left heatmap) and the model’s resulting output intensities are shown in the right heatmap. Rows correspond to Schwartz values, and columns correspond to five preset intensity-combination settings (Sets 1–5).
>
> We will revise the figure captions and main-text explanation accordingly, add clearer axis labels (“X = input intensity,” “Y = measured output ∆”), and visually separate the two panels with distinct titles to improve readability.
>
> ---
>
> **We hope these additional results address your concerns. Thank you again for your thoughtful review and suggestions.**

---

### Official Review · Reviewer_GBae · 2025-10-31

**Soundness:** 3
**Presentation:** 2
**Contribution:** 3
**Rating:** 4
**Confidence:** 4

**Summary:**

VALUEFLOW introduces a unified framework for value-based LLM alignment, addressing extraction, evaluation, and steering gaps. It comprises hierarchical value embeddings unifying SVT, MFT, duties, rights, large-scale intensity-labeled texts via ranking aggregation, and a ranking-based evaluator using VIDB anchors for calibrated intensity scores. Experiments across 10 models reveal steerability asymmetries, multi-value composition laws, and improved demographic prediction on OpinionQA (>10% accuracy gains). The framework enables pluralistic, intensity-controlled alignment.

**Strengths:**

- The ranking-based value evaluation is a novel and timely contribution. It's a promising direction to overcome the reliability and consistency issue of prior evaluation methods.
- Unifying heterogeneous value theories is an interesting and pioneering attempt.
- The large-scale intensity database, upon its open-source, is a significant contribution to the community.

**Weaknesses:**

The methodological section is hard to follow. For example, it is unclear what the motivation and theoretical basis are for the two-stage training process (Section 4.3). How does the unified taxonomy contribute to the value evaluation? What is the relationship between the anchors in Section 4.3 and those in Section 5.2? In Figure 3, how are parts (a) and (b) used together synergistically?

What is the theoretical justification is for using the Plackett–Luce model among all possible scales?

Is the zero-shot, ranking-based value evaluation reliable? Was the evaluator trained? The experiments in Section 3 seem to focus only on SVT values, rather than on all the values used in this study. How do you validate the accuracy of the value evaluation?

How well do your embedding model and value evaluator generalize across different context lengths?

There appears to be no alignment training, despite the claims in the abstract and introduction.

The evaluations in Sections 6.2 and 6.3 have also been conducted in prior work. How does your evaluation improve upon previous ones? Are there any novel insights?

What is the motivation behind designing a unified framework if the components are not trained synergistically? In what way is the end-to-end workflow superior to prior approaches that design individual components separately?

**Questions:**

See above.

---

> ### Author Response · Authors · 2025-11-23
>
> **Dear Reviewer GBae,**
>
> We sincerely appreciate your careful and insightful review. Your comments have been instrumental in improving the clarity and rigor of the paper.
>
> Below we address each point in detail.
>
> ---
>
> ### **W1.1 Motivation and theoretical basis for the two-stage training process**
>
> > For example, it is unclear what the motivation and theoretical basis are for the two-stage training process (Section 4.3). How does the unified taxonomy contribute to the value evaluation?
> >
>
> **Summary**:
> Because Stage 1 learns only within-theory structure, **cross-theory anchor alignment remains weak** (distances are nearly identical: **0.627 vs. 0.678**). Incorporating the unified taxonomy in Stage 2 yields **clear separation**, with cross-theory same-anchor pairs becoming much closer (**0.262 vs. 0.436**).
>
> **Response**:
>
> Thank you for the helpful question. We clarify the motivation and theoretical basis behind the two-stage training process.
>
> **Stage 1** focuses solely on **intra-theory alignment**: the model learns each theory’s hierarchy independently, with no mechanism to link conceptually equivalent values across theories. Because datasets use separate labels and share no cross-theory identifiers, the **model cannot bring semantically aligned texts** **closer** together. As a result, such expressions remain inconsistently placed in the embedding space. Empirically, cross-theory pairs sharing the same anchor (**0.627**) are only slightly closer than generic cross-theory pairs (**0.678**) and even same-theory pairs (**0.701**), indicating that theory boundaries dominate the geometry and cross-theory semantic alignment is largely missing.
>
> **Stage 2** corrects this by incorporating the **unified cross-theory taxonomy** which provides explicit semantic correspondences across heterogeneous theories. After Stage 2, cross-theory same-anchor pairs become substantially closer (**0.262**) than both generic cross-theory pairs (**0.436**) and different-anchor cross-theory pairs (**0.456**), producing a ~43% separation (vs. ~7% in Stage 1). The same improvement holds for individual values (0.334 vs. 0.441).
>
> These results demonstrate that Stage 2 is essential for constructing a coherent, theory-agnostic value-embedding space. We have included this motivations in the revised **Section 4.2~4.3**.
>
> ---
>
> ### **W1.2 Relationship between the two anchors**
>
> > What is the relationship between the anchors in Section 4.3 and those in Section 5.2?
> >
>
> Thank you for pointing this out. The two uses of the term “anchor” indeed refer to different objects, and we understand how this caused confusion. In **Section 4.3**, *anchors* refer to **training-time conceptual anchors**—pooled cross-theory embeddings used to unify heterogeneous taxonomies and regularize the shared embedding space during inter-theory alignment. In contrast, **Section 5.2** uses *anchors* to denote **evaluation-time textual anchors** from the VIDB, which are not involved in training but instead serve as fixed reference texts for estimating intensity through repeated comparisons.
>
> We have updated **Section 5.2** to clearly distinguish these two uses and avoid ambiguity.
>
> ---
>
> ### **W1.3 Figure 3, usage of (a) and (b)**
>
> > In Figure 3, how are parts (a) and (b) used together synergistically?
> >
>
> Thank you for the question.
>
> Parts (a) and (b) are tightly connected: **Figure 3(a)** constructs the **Value Intensity DB (VIDB)**—a calibrated bank of textual anchors with known intensity levels for every value dimension. **Figure 3(b)** then uses this VIDB directly during evaluation: each model-generated response is compared *relatively* against these VIDB anchors to estimate its intensity via ranking and PL optimization. In other words, the VIDB built in (a) becomes the **comparison reference set** used in (b).
>
> We have revised the figure caption (**Figure 3**) to make this dependency more explicit.

---

> ### Author Response · Authors · 2025-11-23
>
> ### **W2. Theoretical justification for Plackett–Luce model**
>
> > What is the theoretical justification is for using the Plackett–Luce model among all possible scales?
> >
>
> **Summary**:
>
> We use the Plackett–Luce family because it smooths cyclic/context-dependent judgments into stable latent utilities and converges far faster than alternatives: with the same comparison budget, PL (w=6) reaches **0.3 mean deviation m=10**, whereas Bradley–Terry and Thurstone remain at **1.3/1.2**.
>
> **Response:**
>
> Our choice of the Plackett–Luce (PL) model follows from the practical objectives of the framework—scalability, stability under large-scale aggregation, and efficient estimation from limited comparisons. Conceptually, value intensity is best represented as a **latent utility score**, and ranking-based models provide a principled way to recover these utilities.
>
> While multiple latent-utility models could be adopted (e.g., Bradley–Terry, Thurstone), PL offered the best trade-off between estimation efficiency and stability in our empirical analysis. In particular:
>
> - For **VIDB construction**, we operate at extremely large scale (≈300K comparisons per value). Using PL with **k=2** (i.e., Bradley–Terry) provides efficient and robust aggregation (**Appendix D.3**).
> - For **evaluation**, where comparisons are limited due to real-time LLM calls, windowed PL with **k=6** converges substantially faster than Bradley–Terry or Thurstone. For example, at **m=3**, PL-6 achieves a mean deviation of **0.8**, compared to **1.7** (BT) and **1.5** (Thurstone) under the same budget (**Appendix D.8**).
>
> More extensive convergence analyses are provided in our **response to** **Reviewer wGtp (W1, Q1)**, to which we kindly refer the reviewer for a detailed discussion.

---

> ### Author Response · Authors · 2025-11-23
>
> ### **W3. Reliability of ranking-based value evaluation**
>
> > Is the zero-shot, ranking-based value evaluation reliable? Was the evaluator trained? The experiments in Section 3 seem to focus only on SVT values, rather than on all the values used in this study. How do you validate the accuracy of the value evaluation?
> >
>
> **Summary**:
>
> We find that the ranking-based evaluator **aligns closely with human judgments** without any training, achieving a **1.4 scalar deviation**, **60–79% win rates** over baselines, **85.3% pairwise accuracy, and 86.7% ±1-window accuracy**, demonstrating robust reliability across all value theories.
>
> **Response**:
>
> We thank the reviewer for highlighting this issue about the reliability of our zero-shot, ranking-based evaluator.
>
> First, we clarify that **the evaluator is not trained**—neither for VIDB construction nor for generation-time evaluation. It operates entirely in a zero-shot ranking mode. Despite the absence of training, we find its behavior to be **highly reliable**, especially when compared against rating-based baselines.
>
> While Section 3 focuses on SVT as the running example, we agree that broader validation is necessary. To address this, we performed a dedicated **human evaluation study** covering value theories used in the paper. In total, we collected **2K scalar ratings** and **1.5K pairwise & windowed ranking judgments** across all values.
>
> We validate the evaluator across three dimensions:
>
> - **VIDB score reliability.**
>
>     We evaluate how closely the evaluator’s scalar intensity scores match human judgments by computing the mean absolute deviation from 2K human-provided scalar ratings and comparing this deviation against rating-based LLM baselines using a win-rate metric that measures how often a model’s score is closer to the human rating.
>
> - **Pairwise ranking accuracy.**
>
>     For sampled text pairs, humans select which text better reflects a target value, and we measure the agreement rate between the evaluator’s ranking and the human choice while also comparing the mean intensity gap for consistent versus inconsistent cases to determine whether disagreements arise from ambiguous pairs.
>
> - **Windowed evaluation fidelity.**
>
>     In 6-window ranking tasks, humans assign each text to one of six ordered windows, and we compute the exact-match accuracy, ±1-window accuracy, and mean positional deviation between human and evaluator assignments.
>
>
> **Table 1. VIDB score reliability.**
>
> | **Ours** | **VS. Qwen3** |  | **VS. Phi4** |  | **VS. Gemma3** |  | **VS. Mistral3.1** |  |
> | --- | --- | --- | --- | --- | --- | --- | --- | --- |
> | Avg. diff | Avg. diff | Win rate | Avg. diff | Win rate | Avg. diff | Win rate | Avg. diff | Win rate |
> | **1.4** | 2.1 | **60.4** | 4.2 | **66.5** | 2.5 | **65.5** | 4.2 | **78.7** |
>
> **Table 2. Pairwise and windowed evaluation.**
>
> | **Pairwise** |  |  | **Windowed** |  |  |
> | --- | --- | --- | --- | --- | --- |
> | Consistency (%) | Mean diff (consistent) | Mean diff (inconsistent) | Exact Acc | ±1 Acc | Mean diff |
> | **85.3** | 6.44 | 2.80 | **60.8** | **86.7** | 0.46 |
>
> Our evaluator demonstrates strong reliability across all measures. For **VIDB score reliability**, it shows the lowest deviation from human scalar ratings (**1.4**) and consistently outperforms rating-based baselines with **60–79%** win rates, indicating closer alignment with human judgments. For **pairwise ranking accuracy**, human–evaluator agreement reaches **85.3%,** and the small gap between consistent and inconsistent cases (2.80 vs. 6.44) suggests that disagreements occur mainly on inherently ambiguous pairs. For **windowed evaluation fidelity**, despite the more challenging **6-window setting**, the evaluator achieves 60.8% exact-match accuracy, 86.7% ±1-window accuracy, and a mean positional deviation below 0.5, demonstrating close agreement with human window assignments.
>
> Together, these results show that our zero-shot ranking evaluator is **consistent with human preferences**, more stable than rating-based baselines, and reliable across multiple value theories. We have added the results to **Section 7.2** and the detailed protocol to **Appendix G**.

---

> ### Author Response · Authors · 2025-11-23
>
> ### **W4. Generalization to different context lengths**
>
> > How well do your embedding model and value evaluator generalize across different context lengths?
> >
>
> **Summary**:
>
> Both components generalize well: **embedding quality stays** **within ±2 points across text lengths**, and **evaluator scores shift by only ≤0.5 across 50–500-word paraphrases**, showing robustness to wide variation in context length.
>
> **Response:**
>
> Thank you for the thoughtful question regarding how well our methods generalize across different context lengths. To address this, we conduct two analyses—one for the embedding model and one for the evaluator.
>
> **1) Embedding Model**
>
> We evaluate whether embedding quality varies with input text length using the same metrics reported in Sec. 6.1: **ranking accuracy** and **similarity correlation**. Texts are grouped into six length buckets, and metrics are computed for each.
>
> **Table 1. Embedding Quality Across Lengths**
>
> | Length Range | Rank Accuracy | Similarity Correlation |
> | --- | --- | --- |
> | < 50 | 69.8 | 39.8 |
> | 50–100 | 69.1 | 37.8 |
> | 100–150 | 71.7 | 38.1 |
> | 150–200 | 70.6 | 39.3 |
> | 200–400 | 71.2 | 41.5 |
> | > 400 | 70.8 | 38.9 |
>
> Across all length buckets, both metrics remain stable, with fluctuations within ±2 points. **Thus, no meaningful degradation is observed across short or long contexts**, indicating that our hierarchical value embedding model generalizes well to diverse text lengths.
>
> **2) Value Evaluator**
>
> To assess length sensitivity in the evaluator, we sampled **1500 generated responses** containing value-related content. For each response, we prompted LLM to produce paraphrases of controlled lengths: **50, 100, 200, and 500 words**. We then fed these paraphrases into our evaluator and computed the score difference relative to the 200-word baseline.
>
> **Table 2. Evaluator Score Shift by Length**
>
> | Length | Mean Δ Score |
> | --- | --- |
> | 50 | –0.52 |
> | 100 | –0.30 |
> | 200 | 0.00 |
> | 500 | +0.51 |
>
> Given the full range (-10~10), the observed score shifts are small in magnitude (≤0.5). We do observe a mild tendency for **longer paraphrases to receive slightly stronger intensity scores**. A plausible interpretation is that longer texts provide more semantic cues that LLM evaluators interpret as stronger evidence for value-relevant intent.
>
> ---
>
> ### **W5. Alignment method used in the paper**
>
> > There appears to be no alignment training, despite the claims in the abstract and introduction.
> >
>
> **Summary**:
> We position alignment within the broader landscape of **inference-time steerable methods**, and our contribution is a framework that **guides models toward target value–intensity profiles** and **evaluates alignment through calibrated measures**.
>
> **Response**:
>
> Thank you for the opportunity to clarify this point.
>
> Our work builds on recent research that expands “alignment” beyond traditional weight-update methods (e.g., RLHF, SFT) to include **training-free, inference-time approaches**. As shown in recent studies [1], alignment can also be achieved by steering model behavior during inference—through prompt-based techniques [2] or activation-level guidance [3]—without modifying model parameters.
>
> Within value-oriented alignment, researchers have likewise emphasized **steerable alignment**, where models are guided toward desired value expressions during inference. Examples include reasoning approaches [4] and profile-based methods [5]. These works collectively show that inference-time methods constitute an established and growing alignment paradigm.
>
> Building on this perspective, our contribution centers on **steerable, pluralistic value alignment** at inference time. VALUEFLOW uses multiple steering channels—including structured prompting, profile-based conditioning, and activation-level variants—to guide model behavior toward target values and intensities. This framing treats alignment not solely as weight modification but as **controlling model behavior** to reflect diverse human values.
>
> In addition, our framework offers a calibrated testbed for evaluating value alignment: using VIDB and our ranking-based evaluator, we measure how well a model moves toward specified value–intensity targets without relying on fixed rating scales.
>
> We will revise the introduction to clarify that our primary contribution is a **steerable, inference-time alignment framework and evaluation protocol**, rather than a new training algorithm.
>
> [1] Pan et al., *A Survey on Training-free Alignment of Large Language Models*, EMNLP 2025.
>
> [2] Cheng et al., *Black-Box Prompt Optimization: Aligning Large Language Models without Model Training*, ACL 2024.
>
> [3] Wang et al., *InferAligner: Inference-Time Alignment for Harmlessness through Cross-Model Guidance*, EMNLP 2024.
>
> [4] Guo et al., *Counterfactual Reasoning for Steerable Pluralistic Value Alignment of LLMs*, NeurIPS 2025.
>
> [5] Sorensen et al.,*Value Profiles for Encoding Human Variation*, EMNLP 2025.

---

> ### Author Response · Authors · 2025-11-23
>
> ### **W6. Evaluation compared to previous ones**
>
> > The evaluations in Sections 6.2 and 6.3 have also been conducted in prior work. How does your evaluation improve upon previous ones? Are there any novel insights?
> >
>
> **Summary**:
> Unlike prior evaluations that rely on fixed rating scales and only assess movement toward a single positive target, our ranking-based, intensity-aware evaluator uniquely enables analysis of **polarity, negative steering, dose–response dynamics, and multi-value effects**, offering insights that existing methods fundamentally cannot measure.
>
> **Response**:
>
> Thank you for this helpful suggestion regarding the scope and novelty of our evaluation.
>
> Although prior work has analyzed LLM values—and noted hard-to-steer dimensions and model-dependent steerability [1–3]—most evaluations target only a single positive direction per value and ignore graded intensity. Without treating **intensity** as a first-class variable, earlier studies cannot capture behaviors such as **negative steerability**, **dose–response curves**, or **nonlinear shifts**. Our novelty lies in introducing **intensity-calibrated steering** together with a **robust ranking-based metric**, which allows us to systematically measure positive vs. negative responses, saturation effects, polarity asymmetries, and multi-value interactions.
>
> This intensity-based evaluation further enables our demographic profile alignment setting: existing methods cannot infer value preferences from **open-ended text** because they lack a mechanism for quantifying **value intensity**, and thus rely on **fixed survey-style rating scales**. In contrast, our framework supports value-based alignment and steerability assessment **without predefined questionnaires**, allowing evaluation directly from open text in a flexible and scalable way.
>
> [1] Ren et al., *ValueBench: Towards Comprehensively Evaluating Value Orientations and Understanding of Large Language Models,* ACL 2024.
>
> [2] Yao et al., *Value FULCRA: Mapping Large Language Models to the Multidimensional Spectrum of Basic Human Values,* NAACL 2024.
>
> [3] Jin et al., *Internal Value Alignment in Large Language Models through Controlled Value Vector Activation*, ACL 2025.
>
> ---
>
> ### **W7. Superiority of end-to-end workflow**
>
> > What is the motivation behind designing a unified framework if the components are not trained synergistically? In what way is the end-to-end workflow superior to prior approaches that design individual components separately?
> >
>
> **Summary**:
>
> Prior work relies on **incompatible theories and scoring schemes**, preventing extraction, evaluation, and steering from working coherently. VALUEFLOW instead offers a **shared intensity axis that unifies all components**, and ablations showing performance drops without each part highlight the strength of this integrated workflow.
>
> **Response**:
> We appreciate the opportunity to clarify this point.
>
> Although VALUEFLOW’s components are not jointly optimized, they are designed to work synergistically through a shared theoretical foundation and an intensity-centered interface. In contrast, prior work typically treats extraction, evaluation, and steering as isolated modules built on incompatible theories and scoring schemes, making integration difficult. Because existing frameworks lack a unified space spanning moral theories, rights, and duties, they cannot mix or compare value signals across theories or support end-to-end workflows requiring consistent cross-theory reasoning.
>
> VALUEFLOW consists of three complementary components—**HIVES** (a unified cross-theory embedding space), **VIDB** (a calibrated intensity scale), and an **ranking-based evaluator** on the same axis. Their synergy stems from a shared theoretical foundation rather than joint training. An ablation on **demographic-profiling alignment** (OpinionQA) shows that removing any component disrupts steering, leading to clear performance degradation.
>
> **Table 3: Ablation on model components**
>
> | Model / Setting | OpinionQA Accuracy (%) |
> | --- | --- |
> | **VALUEFLOW (full)** — Qwen3-32B + HIVES + intensity steering | **58.6** |
> | **w/o intensity steering** | 54.9 |
> | **w/o ranking-based evaluation** | 56.8 |
> | **HIVES → Qwen3-embedding** | 57.1 |
> | **HIVES → UniVar encoder** | 56.0 |
>
> Without intensity-aware steering, accuracy drops; replacing the ranking-based evaluator with a rating model further reduces performance; and substituting HIVES with generic embeddings (Qwen3) or a UniVar removes the cross-theory structure, causing an additional decline. These results show that VALUEFLOW’s coherence is not incidental: each component contributes critically to stable, theory-grounded, and steerable value alignment.
>
> ---
>
> **Please let us know if any additional clarification would be helpful. Thank you again for your detailed and thoughtful feedback.**

---

> > ### Comment · Reviewer_GBae · 2025-11-26
> >
> > Thank you for the response. My concerns have been largely addressed, and I have raised my rating to 6. Given the complexity of the framework, I encourage the authors to further refine the writing to ensure the main content is self-contained and easy to follow.

---

> > > ### Author Response · Authors · 2025-11-26
> > >
> > > Thank you very much for your thoughtful follow-up and for updating your rating. We truly value the attention you have given to our work. We appreciate your concerns regarding the complexity of the framework and the need for greater self-containment in the main text. In the revision, we will clarify the core components of the framework, incorporate key explanations from the rebuttal into both the main paper and the appendix, and strengthen the overall organization so that the narrative is easier to follow.
> > >
> > > Thank you again for your constructive feedback and for helping us improve both the clarity and presentation of the paper.

---

### Author Response · Authors · 2025-12-03
**General Response to Reviewers and ACs**

**Dear Reviewers and Area Chairs,**

We sincerely appreciate the time and thoughtful attention you devoted to reviewing our submission. We are grateful that multiple reviewers highlighted several core strengths of our work: (i) the **ranking-based evaluator and large-scale VIDB** as a meaningful and timely contribution (GBae, Uk4Z); (ii) the ability of **HIVES to unify heterogeneous value theories** into a coherent representation (GBae, Uk4Z, wGtp); (iii) the **comprehensive evaluation**, revealing asymmetric steering and composition effects (wGtp, cFGL); and (iv) the novelty of an **end-to-end framework** connecting extraction, evaluation, and steering (Uk4Z, cFGL).

We have revised the manuscript to clarify methodological components, strengthen the empirical justification for key design choices, and improve coherence across sections. Below, we concisely summarize how we addressed the main categories of concerns raised during the discussion.

---

### **1. Justification for Plackett–Luce Models (Reviewers GBae-W2, wGtp-W1)**

We clarify that the Plackett–Luce (PL) family is appropriate because it aggregates noisy, context-dependent, and occasionally cyclic comparisons into a stable intensity scale. For VIDB, PL integrates ~300K heterogeneous comparisons per value while smoothing IIA violations. Our ablations on the evaluator further show that **PL converges faster than Bradley–Terry or Thurstone** under equal comparison budgets, **justifying its use as the primary aggregation method** (Appendix D.8).

---

### **2. Reliability of the Ranking-Based Evaluator (Reviewers GBae-W2, cFGL-W1)**

We conducted a **human study** with **2K scalar ratings** and **1.5K ranking judgments**, showing strong alignment with humans—**1.4 mean deviation**, **85.3% pairwise accuracy**, **86.7% ±1-window accuracy**—and consistently outperforming rating-based LLM baselines. These results confirm that our ranking-based evaluator provides a stable, human-aligned measure of value intensity (Section 7.2; Appendix G).

---

### **3. Generalization & Additional Experiments**

We additionally conducted several analyses to assess the robustness and breadth of the framework.

**Context-length generalization (Reviewer GBae-W4):** We evaluated HIVES and the evaluator across length buckets and paraphrase lengths and found that embedding metrics and evaluator scores vary little, demonstrating **robustness to variation in context length** (Section 7.2; Appendix D.7).

**Multi-turn conversations (Reviewer wGtp-W2):** We analyzed steering stability in a 100-turn dialogue setup and observed a gradual decay of injected intensities, showing that the evaluator **reliably tracks temporal consistency** (Appendix D.9).

**Generalization (Reviewer wGtp-W2, Q3):** We extended our pipeline to additional **languages** (Chinese, Korean, Arabic) and **distinct value systems** (e.g., Buddhism). By constructing and validating new value items, we demonstrate reliable generalization to both multilingual and non-Western settings (Section 7.2; Appendix F).

---

### **4. Clarifications**

We provided additional clarifications on several methodological and experimental points, including: (i) the **motivation** and necessity of the **two-stage embedding procedure** (Reviewer GBae-W1); (ii) the **rationale for a unified framework** and the novelty of our ranking-based evaluation design (Reviewer GBae-W6, W7); (iii) experimental setup details, including data splits, steering protocols, and evaluation procedures (Reviewers Uk4Z-W2, wGtp-W3, W4); and (iv) the inclusion of **additional baselines**, comparisons, and ablations where appropriate (Reviewer Uk4Z-W1, W3).

---

Several reviewers have acknowledged that **their concerns were resolved and maintained or increased their positive scores** following our responses (e.g., Reviewer GBae: 4→6; Reviewers wGtp and cFGL: maintained 6). Although Reviewer Uk4Z has not yet provided a follow-up, we believe the revised manuscript and clarifications **directly address the key uncertainties** and resolve the major points of misunderstanding raised in the initial reviews.

In summary, the revised submission now presents the VALUEFLOW framework with greater clarity, rigor, and completeness, thanks to the reviewers’ thoughtful comments and constructive feedback.

**Best regards,**

**The Authors**

---

### Meta-Review · Area_Chair_qRAY · 2026-01-08

**Summary:**

Reviewer concerns ranged from evaluation validity. I.e., the reliability of LLM-as-a-judge; soundness of ranking-based intensity modelling raising questions around the suitability of Plackett–Luce, robustness under non-transitivity, and efficiency; VIDB construction consistency concerning apparent contradiction between criticizing rating instability and using llms to label intensity; lack of technical clarity and framework completeness; evaluation scope particularly concerning short-term, english-centric, prompt-only settings; difficult to follow paper presentation and organisation; normative definition of "alignment", as well as dependence on synthetic/prompted settings.

**Reviewer Concerns:**

The paper has massively improved as the authors addressed (some fully, some only partially) the concerns raised by reviewers. The added substantial human evaluation resolves the evaluation validity concern. To address concerns pertaining to soundness of ranking-based intensity modelling, the authors provided theoretical justification for PL as a smoothing latent utility estimator under noisy comparisons and ran ran convergence and ablation studies. The authors acknowledge and clarify wording ambiguities partially addressing the VIDB construction consistency concerns. Another concern pertains lack of technical clarity and framework completeness and efforts including expanded explanations of steering mechanisms and extensive algorithmic descriptions somewhat address this. Regarding evaluation scope concerns, the additional material included detailing new experiments strengths their generality claims but does not fully resolve it. Although authors have indicated that they are committed to clearer structure and smoother language, the paper remains dense, difficult to follow with with complex transitions between extraction, evaluation, and steering. And finally, concerns regarding lack of clarity around the normative definition of "alignment", and dependence on synthetic/prompted settings, although acknowledged by authors, remain unaddressed.

**Reviewer Scores:**

This paper has received Ratings of 1x 4 (marginally below the acceptance threshold), 1 x 2 (reject), and 2 x 6 (marginally above the acceptance threshold)

---

### Decision · Program_Chairs · 2026-01-26

Reject